# Placenta-Derived Extracellular Vesicles in Pregnancy Complications and Prospects on a Liquid Biopsy for Hemoglobin Bart’s Disease

**DOI:** 10.3390/ijms24065658

**Published:** 2023-03-16

**Authors:** Piya Chaemsaithong, Suchaya Luewan, Mana Taweevisit, Wararat Chiangjong, Pisut Pongchaikul, Paul Scott Thorner, Theera Tongsong, Somchai Chutipongtanate

**Affiliations:** 1Department of Obstetrics and Gynecology, Faculty of Medicine, Ramathibodi Hospital, Mahidol University, Bangkok 10400, Thailand; 2Department of Obstetrics and Gynecology, Faculty of Medicine, Chiangmai University, Chiangmai 50200, Thailand; 3Department of Pathology, Faculty of Medicine, Chulalongkorn University, Bangkok 10330, Thailand; 4King Chulalongkorn Memorial Hospital and Thai Red Cross Society, Bangkok 10330, Thailand; 5Pediatric Translational Research Unit, Department of Pediatrics, Faculty of Medicine Ramathibodi Hospital, Mahidol University, Bangkok 10400, Thailand; 6Chakri Naruebodindra Medical Institute, Faculty of Medicine Ramathibodi Hospital, Mahidol University, Samut Prakan 10540, Thailand; 7Integrative Computational BioScience Center, Mahidol University, Nakhon Pathom 73170, Thailand; 8Institute of Infection, Veterinary and Ecological Sciences, University of Liverpool, Liverpool CH64 7TE, UK; 9Department of Laboratory Medicine and Pathobiology, University of Toronto, Toronto, ON M5S1A8, Canada; 10Division of Epidemiology, Department of Environmental and Public Health Sciences, University of Cincinnati College of Medicine, Cincinnati, OH 45267, USA

**Keywords:** biomarkers, diagnosis, exosomes, hemoglobinopathy, hydrop fetalis, liquid biopsy, placental hypoxia, thalassemia

## Abstract

Extracellular vesicles (EVs) are nano-scaled vesicles released from all cell types into extracellular fluids and specifically contain signature molecules of the original cells and tissues, including the placenta. Placenta-derived EVs can be detected in maternal circulation at as early as six weeks of gestation, and their release can be triggered by the oxygen level and glucose concentration. Placental-associated complications such as preeclampsia, fetal growth restriction, and gestational diabetes have alterations in placenta-derived EVs in maternal plasma, and this can be used as a liquid biopsy for the diagnosis, prediction, and monitoring of such pregnancy complications. Alpha-thalassemia major (“homozygous alpha-thalassemia-1”) or hemoglobin Bart’s disease is the most severe form of thalassemia disease, and this condition is lethal for the fetus. Women with Bart’s hydrops fetalis demonstrate signs of placental hypoxia and placentomegaly, thereby placenta-derived EVs provide an opportunity for a non-invasive liquid biopsy of this lethal condition. In this article, we introduced clinical features and current diagnostic markers of Bart’s hydrops fetalis, extensively summarize the characteristics and biology of placenta-derived EVs, and discuss the challenges and opportunities of placenta-derived EVs as part of diagnostic tests for placental complications focusing on Bart’s hydrop fetalis.

## 1. Introduction

Thalassemia is the most common hematologic genetic disease in Southeast Asia. Alpha-thalassemia major (homozygous alpha-thalassemia-1) or hemoglobin (Hb) Bart’s disease is the most severe form of thalassemia disease [1,2,3,4,5]. The prevalence of Hb Bart’s disease is approximately 0.23% [6], with a deletion frequency in Southeast Asia as high as 4.5–5% [7]. The term Hb Bart’s hydrops fetalis was first described in 1960 [8] and the fetuses with this condition are characterized by severe anemia, hypoxia, heart failure, and hydrops fetalis (accumulation of body fluid) [9]. Unfortunately, these fetuses usually die in utero or in the early neonatal period, although the survivors after gene therapy or bone marrow transplantation have been reported [10,11,12]. These survivors remain transfusion dependent. 

Molecular diagnosis of Hb Bart’s hydrops fetalis is characterized by homozygous α-thalassemia-1 (-/-); therefore, the entire delta and globin gene clusters are lacking but present only Hb Bart’s (γ4) [1]. Such Hb has a high oxygen affinity, creating widespread tissue hypoxia and fetal anemia. Hemodynamic changes in fetal anemia are cardiomyopathy, increased intravascular volume, hepatic extramedullary hematopoiesis, and widespread endothelial cell damage resulting in cardiomegaly, hepatosplenomegaly, and placentomegaly [13,14]. Currently, prenatal screening of this condition is possibly performed by the determination of the mean corpuscular volume, dichlorophenolindophenol, osmotic fragility, or Hb typing [15,16]. Any couple at risk of Hb Bart’s hydrops fetalis is counseled to perform an invasive diagnostic test (chorionic villus sampling, amniocentesis, or cordocentesis) or to follow up with a non-invasive approach by performing ultrasonography to evaluate the cardiac diameter/thoracic ratio (CTR), middle cerebral artery peak systolic velocity (MCA PSV), or placental thickness [17,18,19,20,21]. Maternal blood biomarkers including non-invasive prenatal diagnosis tests such as prenatal cell-free fetal DNA for the identification of Hb Bart’s hydrops fetalis are poor predictors, thus, they are not routinely used in a clinical setting [22,23,24,25,26,27].

With a better understanding of intercellular communication, the role of extracellular vesicles (EVs) has emerged [28,29,30,31,32]. Accumulating evidence has shown that placenta-derived EVs can be identified in maternal plasma at as early as six weeks of gestation and their concentrations have increased as a function of gestation in normal pregnancy [33]. Maternal plasma concentrations of placenta-derived EVs released by the syncytiotrophoblast are further increased in pregnancy complications associated with placental hypoxia [34,35,36,37]. In this article, we will summarize the current evidence of biomarkers for the identification of Hb Bart’s hydrops fetalis, a condition associated with placental hypoxia [38], and discuss the challenges and opportunities of clinical implications of placenta-derived EVs for this lethal disease.

## 2. Bart’s Hydrops Fetalis: Clinical Perspectives and Current Diagnostic Procedures

In Hb Bart’s disease, the placenta can range from near normal in appearance to enlarged, friable, pale, and edematous (hydropic) [39,40] (Figure 1a). Their weights are often excessive, as much as 2 kg [21]. These changes reflect the adaptive placental response to severe and prolonged intrauterine hypoxia, beginning as early as the first trimester of pregnancy. The placental changes in Hb Bart’s disease involve both villous maturation and vascularity [41,42,43].

In the normal placenta, mesenchymal villi first develop at 5 weeks’ of gestation, serving as the precursors to more mature villous types. Around 8 weeks of gestation, mesenchymal villi develop into immature intermediate villi, which become the predominant type by 14–20 weeks of gestation. As the pregnancy proceeds, immature intermediate villi transform into stem villi. Throughout the third trimester, mesenchymal villi differentiate into mature intermediate villi, which then later produce terminal villi growing in grape-like clusters as the final stage of villous development [44]. The placenta in Hb Bart’s hydrops increases the number of immature intermediate villi, which persist despite advancing in gestational age, and this finding is referred to as “generalized delayed villous maturation” [38,40]. Although Hb Bart’s fetuses suffer from severe anemia, the survival in the embryonic period is uneventful because of the production of the embryonic Hbs Portland I (ζ2γ2) and Gower I (ζ2ε2), which maintain the Fetal-placental circulation [45]. The switching from embryonic to abnormal fetal Hb (Hb Bart’s) that occurs around 6–8 weeks of gestation is concurrent with the transition of mesenchymal villi to immature intermediate villi [19,21,40]. This switch to an abnormal Hb may initiate or aggravate the increase in the number of immature intermediate villi, resulting in the persistence of such villi throughout gestation. 

Other villous changes in the Hb Bart’s placenta involve the villous stroma and the cytotrophoblastic cells that cover the villous stroma. Cytotrophoblastic cells tend to be numerous and conspicuous in the Hb Bart’s placenta throughout gestation, in contrast to placentas in uncomplicated pregnancies where these cells become scarce by the third trimester [40]. Chronic hypoxia stimulates the proliferation of such cells but inhibits their fusion and differentiation into syncytiotrophoblast [42]. Within the stroma of the immature and mature intermediate villi of the Hb Bart’s placenta, there is often a characteristic increase in the number of stromal cells at the periphery beneath the trophoblast layer, a finding referred to as “peripheral villous stromal hypercellularity (PVSH)” [40] (Figure 1b). This change is seen in >80% of Hb Bart’s cases and is usually multifocal. PVSH is hypothesized to be a placental adaptation to chronic hypoxia, in utero. The immature intermediate villi have a large diameter, and their predominance in the Hb Bart’s placenta leads to the narrowing of the intervillous space, impeding fetoplacental oxygen and nutrition transfer. Contraction of the myofibroblastic cells in PVSH would serve to reduce the villous size, thereby widening the intervillous space for an increase in the maternal blood flow.

Vascular formation is a crucial step in placental development, and this is influenced by numerous factors, one of which is chronic hypoxia [21,41,42]. The pattern of the placental vascular adaptive response is influenced by the source of chronic hypoxia and can be classified into pre-placental (maternal), placental, and post-placental (fetal) sources. Pre-placental causes of hypoxia include maternal anemia, maternal diabetes, or maternal smoking, whereas placental causes result from a restricted supply of normoxic maternal blood to the placenta, as seen with preeclampsia at term and fetal growth restriction with preserved end-diastolic flow in umbilical arteries. In these situations, the oxygen level of the intervillous space is low and this stimulates a predominantly branching pattern of villous vascularization. On the other hand, in post-placental hypoxia, there is a failure of the fetoplacental extraction of oxygen from the intervillous space. Post-placental hypoxia can be observed in stillbirth or fetal growth restriction with an absent or reverse end-diastolic umbilical blood flow. The relatively high oxygen level in the intervillous space stimulates a predominantly non-branching pattern of villous vascularization. An increased number of villous vessels can be seen in pre-placental, placental, and post-placental hypoxia (Figure 1c).

Patterns of placental vasculature can be accurately assessed with the aid of computer-assisted measurements on placental histologic sections [38]. In Hb Bart’s, the number of villous vessels in the placenta is markedly increased. There is also a significant increase in the vascular perimeter and endothelial thickness. The etiology of the hypoxia-induced placental vascularization is multifactorial in Hb Bart’s disease. Morphometric studies have identified a branching vascular pattern, a change that is associated with pre-placental and placental hypoxia. This change is thought to have its basis for the marked placental enlargement, which compromised the blood flow, from the uterine distention and the generally diminished intervillous space due to the numerous intermediate types of villi. However, post-placental hypoxia is also likely to be in effect, given the greatly reduced capacity of Hb Bart’s to extract oxygen from the intervillous space.

### Current Biomarkers for Hemoglobin Bart’s Disease

The non-invasive prenatal diagnosis test (NIPD) for fetal Hb Bart’s disease utilizes sonographic and maternal serum biomarkers as screening tools. Nonetheless, in actual practice, only sonographic markers are available, whereas serum biomarkers are rarely used in practice. Sonographic markers can accurately predict fetal Hb Bart’s disease as early as the late first trimester (12–15 weeks of gestation), with a detection rate of 90–95 percent [46,47,48,49]. Almost all cases of fetal Hb Bart’s disease are detected at gestational age (GA) 18–22 weeks. Leng et al. showed that the overall sensitivity and specificity of sonographic markers throughout the gestational period are 100 percent and 95.6 percent, respectively [50]. The potential sonographic markers for predicting fetal Hb Bart’s disease [48,49,51,52,53] include the following: a cardiac diameter/thoracic diameter ratio (CTR) with a cut-off value of 0.5 at mid-pregnancy;a high peak systolic velocity of the middle cerebral artery (MCA-PSV) with a cut-off value of 1.5 multiple of median (MoM), and;an increase in the placental thickness with a cut off value of 1.8 cm in the late first trimester and 3 cm at mid-pregnancy.

Additionally, several other sonographic markers can also support the diagnosis of fetal Hb Bart’s disease, including nuchal translucency, heart circumference, liver length, splenic circumference, and splenic artery peak systolic velocity [49,54,55,56,57]. The performance of sonographic markers is shown in Table 1. Serial ultrasound screening for Hb Bart’s disease during pregnancy, beginning in the first trimester and continuing every 2–4 weeks until 24 weeks, has a sensitivity of 100 percent in detecting pre-hydropic signs and a false positive rate of 10.9 percent. It also reduces the rate of invasive procedures by 70 percent. Hence, in normal practice, the sonographic marker is recommended for the routine screening of fetal Hb Bart’s disease [58]. The main limitations of serial ultrasound are that it requires specific equipment and it is operator-dependent. 

The studies of maternal serum biomarkers, as a part of the second trimester Down syndrome screening [gestational age (GA) 18–22 weeks], demonstrated that serum biomarkers could also predict Hb Bart’s disease. Several maternal serum biomarkers used in routine fetal Down syndrome screening, such as free beta-human chorionic gonadotropin (β-hCG), inhibin-A, pregnancy-associated plasma protein-A (PAPP-A), alpha-fetoprotein (MAFP), and unconjugated estriol (uE3), are probably useful in predicting fetal Hb Bart’s disease. Increased β-hCG, inhibin-A, and PAPP-A levels have been noted from the increased placental hormone synthesis due to placentomegaly in fetal Hb Bart’s disease [22]. Additionally, elevated MAFP and decreased uE3 levels are observed because of hepatomegaly caused by extramedullary hematopoiesis secondary to fetal anemia [64]. According to the study conducted by Wanapirak et al. in 2018 [27], the most significant single predictor is MAFP with a cut-off greater than 1.5 MoM. (The sensitivity of 87.2 percent and the specificity of 74.5 percent). The combination prediction model including AFP and uE3 provided the best diagnostic performance. With the probability cut-off point greater than 0.5, the model gave an acceptable sensitivity of 61.5 percent and a high specificity of 98.1 percent [23]. However, using serum markers as a single tool in screening for fetal Hb Bart’s is not satisfactory. These markers are primarily used to screen for Down syndrome and neural tube defects in fetuses. Moreover, fetal anemia secondary to any causes other than Hb Bart’s disease, such as alloimmunization or Parvovirus B19, can result in an unexpected increase in the MAFP levels. However, the clinicians should take advantage of the second-trimester maternal serum screening to simultaneously screen for fetal Hb Bart’s disease, especially in geographical areas of high prevalence. Combining serum markers (MAFP) with sonographic markers such as CTR, placental thickness, and MCA-PSV may be beneficial with a sensitivity of approximately 48–69 percent and a specificity of 88.9–100 percent [64], as shown in Table 2. Similarly, the placental growth factor (PlGF) and soluble fms-like tyrosine kinase-1 (sFlt-1) commonly used in predicting preeclampsia were also demonstrated to be useful in screening for fetal Hb Bart’s disease. Thus, in fetal Hb Bart’s disease, a significant increase in PlGF and a decrease in the sFlt-1/PlGF ratio are observed, although sFlt-1 is marginally enhanced (*p* values of 0.008, 0.139, and 0.001 respectively) [24].

The novel technology of NIPD using maternal plasma cell-free fetal DNA (cffDNA), has been developed and is now utilized to screen common fetal aneuploidies, microdeletion/ microduplication syndromes, and monogenic disorders after 10 weeks of gestation. In thalassemia, significant progress has been achieved in developing NIPD as an effective screening method. The challenge for development is separating cffDNA from maternal DNA. Several techniques such as reverse transcription-quantitative PCR (RT-qPCR) using the gap-PCR technique, droplet digital PCR (ddPCR), and next-generation sequencing (NGS) have been developed for NIPD [6,65,66,67,68,69,70,71]. The NGS techniques in particular, are highly sensitive and enable the detection of small amounts of cffDNA. According to a recent study involving 878 cases at risk of fetal Hb Bart’s disease, the sensitivity and specificity in detecting fetal Hb Bart’s disease as early as 11–13 weeks of gestation were 98.98 percent and 96.06 percent, respectively [6]. However, the high costs, complex techniques, and unavailability in clinical laboratories make them inappropriate to use in routine clinical practices. In addition, this technique requires further validation. Thus, the search for other biomarkers is continuing. Table 3 demonstrates the advantages and disadvantages of different biomarkers for the identification of Hb Bart’s hydrops fetalis. 

## 3. Extracellular Vesicles (EVs) as a Source of Non-Invasive Liquid Biopsy

EVs are membrane-bound vesicles containing cytosol from secreting cells enclosed in lipid bilayer structures, released into the extracellular environment from all cell types including trophoblasts [72], erythrocytes [73], and endothelial cells [74]. They carry specific cargos containing lipids, proteins, and nucleic acids (i.e., mRNAs, microRNAs, long non-coding RNA, and DNA fragments). EVs play a crucial role in cell-to-cell communication including fetal–maternal communication by the regulation of several biological processes, mediated by their surface receptors and their contents [29,36,75,76,77,78,79,80,81,82]. The concept of EVs was first introduced in 1860s by Charles Darwin [83] who proposed the theory of Pangenesis to understand the mechanisms of inheritance and the cause of natural variation [84]. Other than cell division for transferring genetic information, every cell in the body can produce particles containing diverse molecules called “gemmules” traversing to other cell types [83,84]. The first evidence of cell trafficking between the mother and the fetus was reported by Georg Schmorl in 1893 who demonstrated the fetal cells in the form of thrombin containing multinucleated syncytial giant cells in the lungs of pregnant women with eclampsia [85,86]. In the context of EV, placenta-derived EVs enter target cells in the uterus by endocytosis and are trafficked to early and late endosomes [87]. This pathway might allow for the trafficking of EVs across the syncytiotrophoblast and their release on the opposite surface from multivesicular bodies. Both the feto-maternal and maternal−fetal trafficking of exosomes during pregnancy can be demonstrated by the use of fetal cell-derived fluorescently labeled exosomes or genetically engineered mice, in which fetal and maternal exosomes could be distinguished [88,89]. The detection of fetal exosomes in maternal plasma discloses their potential as biomarkers for pregnancy monitoring using minimally invasive liquid biopsy.

### 3.1. Extracellular Vesicle Subpopulations

EVs are classified into three subpopulations, i.e., exosomes, microvesicles, and apoptotic bodies, based on their sizes, molecular markers, and contents [28]. Figure 2 shows EV biogenesis and molecular compositions. Exosomes are originated by the inward budding of the plasma membrane (endosomal route) forming multivesicular bodies which internalize to generate intraluminal vesicles subsequently released into the extracellular environment as exosomes [90,91,92]. The term intraluminal vesicles is used when the vesicles are inside the multivesicular bodies. In contrast, exosomes are described when the contents of the multivesicular bodies are released into the extracellular environment by cell membrane exocytosis [93]. Typically, exosomes can be distinguished from microvesicles and apoptotic bodies by their size, origination, formation, isolation methods, and protein enrichment as demonstrated in Table 4. To avoid confusion of the EV nomenclature in previous publications, the general term of EV is applied for exosomes and microvesicles (unless specified) throughout this review.

It is now well-established that EVs contain a variety of proteins, lipids, various RNA species (mRNA, microRNA, or other non-coding RNAs), and small amount of DNA as well as putative surface proteins or ligands that bind to receptors on target cells. The cargo from EVs has a distinct molecular signature reflecting the originating cells which facilitate the selective incorporation and triggering of biochemical and/or phenotypic changes in the recipient cells [28,94,95,96]. The cargo contents can be absorbed when EVs circulate, and this leads to the modification of the target gene expression, signal, and biological function of the recipient cells [97]. The expression of exosomal cargos is altered according to the microenvironment [98,99,100].

### 3.2. EV Isolation and Characterization

Several approaches have been applied to isolate EVs, even though there is no universal method that is suitable to all projects [101]. EV isolation based on a single step of polymer/precipitation agents (e.g., polyethylene glycol), high-speed ultracentrifugation, or low-molecular weight centrifugal filters provides the highest amount of EV recovered from the biofluids. However, they come with a trade-off in the specificity due to the significant contamination of vesicular (i.e., microvesicles, apoptotic bodies) and non-vesicular (e.g., free and precipitated proteins, lipoprotein micelles) components [101]. The single step EV isolation by high-molecular weight centrifugal filters, size-exclusion chromatography, tangential flow filtration, or membrane affinity columns, as well as the differential ultracentrifugation with the low/intermediate centrifuge prior to the high-speed, shows an improved specificity while maintaining the intermediate recovery yield [101,102,103,104]. The main advantage of these single-step protocols is the turnaround time and the compatibility with large-scale, exosome-molecular studies, in which many samples have to be processed for EV isolation within a relatively short period of time. 

Combinatorial EV isolation based on two methods (e.g., differential ultracentrifugation coupled with ethylene glycol precipitation, size-exclusion chromatography) [105,106], the immunoaffinity isolation based on the specific EV molecules [107,108], or three combined methods, i.e., step-wise centrifugation, ultrafiltration, and size exclusion chromatography [109,110], usually provide better EV specificity and thus are suitable when the specific goals involve the EV functions or subpopulations. The combined isolation methods are very useful when the known contaminants are expected to significantly interfere with the downstream analysis, i.e., the measuring biomarkers could be presented as free soluble proteins and EV-containing proteins. Note that the multi-step combinatorial protocols and immunoaffinity procedures provide a low recovery yield with as few contaminates as possible [101], so that the downstream analysis and data interpretation can be straightforward with high confidence. However, the reproducibility and the upscaling of the isolated EVs for the whole project should be addressed at the initial phase of the studies. Therefore, one can choose the method by considering the degrees of recovery and specificity of the isolated EVs and their compatibility with the downstream applications [101].

Once isolated, EV-enriched specimens are required to be validated for the EV presence following the International Society for Extracellular Vesicles (ISEV) guideline [101]. In general, EV characterization must be performed to demonstrate both the particle and protein evidence. Particle evidence usually requires two different but complementary methods to be satisfied [101]. These include one technique for high-resolution images of the EV morphology (e.g., transmission electron microscopy or atomic force microscopy) and another technique for the EV size distribution (i.e., nanoparticle tracking analysis, high-resolution flow cytometry, or asymmetric-flow field-flow fractionation) [111,112,113,114]. Protein evidence could be satisfied by demonstrating three positive (enriched) and one negative (depleted) EV marker as follows: (i) the enrichment of plasma membrane-associated EV markers (e.g., CD9, CD63, CD81, MHC molecules, integrins, LAMP1/2, syndecans); (ii) the enrichment of cytosolic proteins involved in EV biogenesis or with a lipid/membrane protein-binding ability (e.g., TSG101, ALIX, flotillin-1 and -2, caveolins, annexins); (iii) the depletion of proteins commonly found as EV contaminants (i.e., free soluble proteins such as lipoproteins, serum albumin, and cytokines) or subcellular compartment proteins (e.g., histones, cytochrome C, calnexin, and HSP90B1) depending on the matrix [101]. Immunodetection methods such as Western blotting and ELISA are often used to show the EV protein evidence and in some circumstances, quantitative proteomics can be applied to identify and quantify multiple EV protein markers to support the claim of EV presence in the isolates [115,116]. Figure 3 demonstrates the commonly used methods for the detection of EVs as well as their cargos.

### 3.3. Placenta-Derived EVs

In pregnancy, the discovery of circulating fetal genetic material in the maternal plasma has enhanced the non-invasive prenatal diagnosis [117,118,119,120,121,122,123,124,125,126]. The placenta secretes a large number of EVs into maternal circulation since they are shed from the syncytiotrophoblast into the intervillous space and then flushed via the uterine veins into the maternal circulation [127]. The placental EVs were then confirmed that they are of placental origin by using the placental alkaline phosphatase (PLAP) marker. In addition, such fragments are detected in all pregnant women, but not in the non-pregnant population, suggesting a pregnancy-specific marker. The placenta-derived exosome can induce apoptosis and the down regulation of T cell responses contributing to fetal immune tolerance [128]. Subsequently, other investigators have also isolated placenta-derived EVs using human placenta explant cultures from the first trimester of normal pregnancies and reported that PLAP is a marker of placental exosomes [129,130,131,132]. Although 87% homology is found with the intestinal alkaline phosphatase gene, there are differences at their carboxyl terminal end [133,134], which is unique to the placenta.

Accumulating evidence shows that placenta-derived EVs exhibit multiple functions, including the following.

#### 3.3.1. Regulating Trophoblast Migration and Angiogenesis

Placenta-derived EVs can regulate trophoblast migration and angiogenesis [72,98,99,135]. In in vitro studies, placenta-derived EVs isolated from pregnant women are biologically active since they can induce endothelial cell migration [132]. In addition, the release of EVs from cytotrophoblast culture depends on the oxygen tension whereby the low oxygen tension environment promotes extravillous trophoblast (EVT) cell migration, endovascular invasion, and proliferation. Thus, it is possible that placenta-derived EVs are involved in the process of placentation [98,136].

#### 3.3.2. Promoting Maternal–Fetal Immune Tolerance

Placenta-derived EVs from in vitro-cultured placental explants down-regulated the NK cell receptor NKG2D and impaired the cytotoxic function of T cells from the peripheral blood mononuclear cells of pregnant women [130,137,138]. Furthermore, placenta-derived EVs suppressed T cell signaling components and induced lymphocyte apoptosis leading to immune tolerance [139]. Despite the immunosuppressive activity of placenta-derived EVs, the risk of infection caused by the EVs remains unremarkable.

#### 3.3.3. Mediating Maternal–Fetal Intercommunication

Placenta-derived EVs play a crucial role in the crosstalk between the feto–placental unit or maternal–fetal communication [36,82,140]. Studies demonstrated that placental-specific microRNAs within C19MC-derived miRNAs can be detected in the maternal circulation via EVs where they function in placental–maternal signaling [141,142]. For example, placenta-derived EVs can be taken up by endothelial cells, immune cells, or platelets and they can modulate their functions such as generating pro-inflammatory signals, inducing death ligands, mediating nitric oxide signaling, or producing damage-associated molecular pattern (DAMP) molecules [143,144,145,146]. In addition, non-placental cells incubated with placenta-derived EVs harboring C19MC miRNA clusters attenuate the viral replication in recipient cells through the induction of the autophagy pathway [142]. Therefore, EVs regulate the maternal immune response to maintain a normal pregnancy and protect against viral infections such as Cytomegalovirus infection [143,147,148].

#### 3.3.4. Regulating Maternal Metabolic Homeostasis

Placenta-derived EVs regulate maternal metabolic homeostasis by transferring the gene information to target cells [29]. For example, miRNAs derived from the placenta-derived EVs of women with gestational diabetes have been shown to reduce insulin sensitivity and glucose uptake by striated muscle [100,149,150,151,152].

Collectively, placenta-derived EVs are specifically packaged with signaling molecules including protein, mRNA, microRNA, and non-coding RNA. They play a role in the translational activities of angiogenesis, immunomodulation, cell signaling, and metabolic homeostasis. As a result, placental exosome signaling represents a fundamental pathway mediating the maternal–fetal intercellular communication [153,154]. 

### 3.4. Pathophysiological Roles of Placenta-Derived EVs in Pregnancy Complications

Placenta-derived EVs have a unique feature due to the presence of specific placental alkaline phosphatase (PLAP) [129,132,139], HLA-G [155], and miRNAs such as those within the chromosome 19 miRNA cluster [75,142,156]. They can be identified by size, buoyant density, and the presence of CD63 as well as PLAP antigens. Placenta-derived EVs are identified in maternal plasma as early as sixth weeks of gestation [33] and the concentration of placenta-derived EVs in maternal plasma increases progressively as a function of gestation [33,132]. Furthermore, the concentration of maternal plasma placenta-derived EVs is approximately 20-fold higher than that observed in non-pregnant women, indicating a pregnancy-specific protein [132,139]. It has been hypothesized that placental and fetal communication as well as the presence of fetal cells in maternal circulation occur beginning from the fourth week of gestation, although intervillous circulation has not been completely established [157].

Several factors such as hypoxia, high glucose concentrations, and inflammatory signals trigger the release of placenta-derived EVs [98,100,158]. A high glucose concentration (25 mM) significantly increased the number of EVs from the first trimester primary trophoblast cells and such EVs triggered the release of pro-inflammatory cytokines from human umbilical vein endothelial cells [100]. A similar observation has been demonstrated in pregnant women with gestational diabetes mellitus [149,150,151]. Specifically, women with gestational diabetes have a higher maternal plasma-derived exosome concentration than those with a normal pregnancy from the 11th week of gestation onwards [150]. EVs isolated from the plasma of women with gestational diabetes induce the release of pro-inflammatory cytokines including granulocyte-macrophage colony-stimulating factor (GM-CSF), interleukin (IL)-4, IL-6, IL-8, interferon-gamma (IFN-γ), and tumor necrosis factor-alpha (TNF-α) [150]. Thus, it is possible that the dysregulation of EVs may be involved in the pro-inflammatory status in gestational diabetes. 

In in vitro studies, the number of EVs released from extravillous trophoblast (EVT) cells is higher than that from EVT cultured under a hypoxic condition (1% oxygen) compared to normal oxygenation (8% oxygen) [158]. Interestingly, EVs isolated from EVT cultured at 1% oxygen reduce the endothelial cell migration but increase the production of TNF-α. Hypoxic conditions also influence exosome contents as well as their biological activities, such as protein and miRNA, against target cells [98,99,158]. These findings have important clinical implications since the determination of the placenta-derived exosome concentration and their cargo contents in maternal plasma can be used as a non-invasive method for the early detection and diagnosis as well as monitoring of placental-associated complications [159]. In modern clinical obstetrics, the term “liquid biopsy” refers to the molecular diagnosis or biomarker determination in biological fluids, especially blood, for the evaluation of pregnancy-related complications or prenatal screenings. Compelling evidence demonstrated that the aberration exosome number and their contents are observed in women with preeclampsia [29,34,37,160], diabetes [35,150,161,162,163], fetal growth restriction [164], and preterm birth [116,165,166,167,168]. Figure 4 shows the role of placental debris and EVs in placental-associated complications. 

Several studies have demonstrated that placental-specific EVs are implicated in the pathogenesis of preeclampsia since they generate inflammatory responses, since they generate inflammatory responses, increase anti-angiogenesis, activate platelets, and induce endothelial cell activation. In an experimental study, syncytiotrophoblast EVs from explant cultures of a preeclamptic placenta induced the release of proinflammatory cytokines such as IL-1β, IL-6, IL-18, macrophage inhibitory protein-1-α and -β, or TNF-α in peripheral mononuclear cells [169]. In addition, the administration of EVs to pregnant mice induces embryonic death or a small size of the embryo and placenta, as well as preeclampsia-like features including elevated blood pressure, proteinuria, enlarged glomeruli, thickened glomerular basement membrane, and increased maternal plasma sFlt-1 concentrations [170]. These changes were not observed in non-pregnant mice. The mechanism responsible for the development of preeclampsia is attributed to the activation of maternal platelets in the placental vascular bed, induced via an adenosine triphosphate (ATP) release and inflammasome activation [170]. In humans, the evidence has demonstrated an alteration in the EVs and their cargos in the placenta and maternal plasma of women with preeclampsia [29,34,37,160]. However, an alteration in EVs in the placenta or maternal circulation can be found in other great obstetrical syndromes such as fetal growth restriction [164] and gestational diabetes [163]. Figure 5 illustrates a possible role of placenta-derived exosomes in Bart’s hydrops fetalis and in the pathogenesis of mirror syndrome. 

## 4. Placenta-Derived EVs as Hb Bart’s Liquid Biopsy: Challenges and Future Perspectives

While placenta-derived EVs are promising as the source of candidate biomarkers with clinical relevance, there are several challenges when conducting EV biomarker studies. As aforementioned, one needs to ensure the performance and compatibility of the EV isolation method chosen to support the specific goals/downstream applications [101]. It should be emphasized that different isolation methods could impact the downstream EV molecular analysis [171], while it is common that the isolated EV samples fail the quality control experiments. Nonetheless, this challenge can be simply addressed by the practice of reproducible EV isolation and some preliminary experiments to choose the suitable isolation method for achieving the qualified EV samples for downstream analyses. In addition, EVs presented in biofluids are heterogeneous in nature [172]. For example, plasma EVs are contributed from the local vascular endothelial cells and distant organ systems (e.g., brain, heart, lung, liver, intestine, and kidneys) and they can change in both quantity and quality during the healthy and disease states. The key to address this issue relies on the study design that focuses on the inclusion and exclusion criteria, the clinical context of using the candidate EV biomarkers, and a sufficient study population to support the statistical power. In addition, the validation phase of biomarker candidates should be performed with independent larger cohorts or multi-center studies [173]. Alternatively, the application of tissue-specific markers, for example, targeting PLAP for the placenta-derived EV subtype [29], is a potential approach to minimize the EV heterogeneity caused by the unrelated organ systems. The multiplex detection of PLAP-positive EVs in relation to other EV molecular profiles (i.e., proteins and miRNAs) should be pursued in future studies involving biomarker discovery and validation for placental-associated complications.

Only a few studies have described exosomes or microvesicles from hemoglobinopathy adult patients [174,175,176,177,178,179]. In addition, most of these studies examined the role of microparticles in circulation from adults with Beta-thalassemia/Hb E disease in relation to the hypercoagulable state. The authors demonstrated that microparticles obtained from Beta-thalassemia/Hb E patients can induce an inflammatory response, endothelial dysfunction [174], and stimulate platelet activation as well as aggregation leading to thrombus formation [175]. Thus far, there is no study determining placenta-derived EVs in pregnant women with Hb Bart’s hydrops fetalis. Since placental hypoxia is a main feature of Bart’s hydropic fetus, it can trigger the release of placental debris or placental-derived exosomes which subsequently enter maternal circulation. In addition, in some cases of Bart’s hydrops fetalis, Ballantyne or mirror syndrome characterized by a simultaneous edematous state of the mother, fetus, and placenta (also called triple edema) can be observed. In this syndrome, the mother may develop proteinuria, hypertension, and even severe preeclampsia [180,181,182].

Altogether, we hypothesize that the placenta-derived exosome concentration might have a role in the identification of women with Hb Bart’s fetuses beginning in early gestation and this might be implicated in the genesis of mirror syndrome in some cases. Ultimately, we propose that the placenta-derived exosome might be used as a liquid biopsy tool for the identification of fetuses at risk for this condition and this might reduce the rate of unnecessary invasive prenatal diagnosis determinations. In this direction, an integrative strategy has been proposed to combine current biomarkers with placenta-derived EVs for the identification of Bart’s hydrops fetalis throughout gestation (Figure 6). Importantly, prior to clinical use, it is necessary to standardize the protocols of exosome isolation and characterization, while the diagnostic performance of placenta-derived EVs should be evaluated as a solitary biomarker and as part of the combined biomarkers in the large-scale prospective cohorts or multicenter studies.

Last but not least, several drugs are known to modulate the EV release and cargo contents, and thus are actively investigated to harness EV-modulated therapeutic effects against non-communicable and infectious diseases [183,184,185]. Among these medications, antiplatelets are commonly used during pregnancy. Low-dose aspirin prophylaxis is indicated in patients at a high-risk of developing preeclampsia [186]. Dual antiplatelet therapy (aspirin and P2Y12 receptor blocker) has been used to manage pregnancy-associated myocardial infarction following a percutaneous coronary intervention (PCI) with intracoronary stenting [187,188,189]. Interestingly, aspirin suppressed the cargo levels of proinflammatory cytokines and mediators in platelet-derived exosomes [190]. Ticagrelor, a potent P2Y12 receptor blocker, induced the growth of cardiac-derived mesenchymal stem cells and enhanced the release of anti-hypoxic exosomes with cardioprotective effects [191]. While future investigations of placenta-derived EVs as the liquid biopsy should take into account the prescribed medications during pregnancy, one can foresee a translational potential of EV modulatory agents, especially antiplatelets, to mitigate the pathophysiological consequences of placental hypoxia and pregnancy complications through the modulations of placenta-derived EV biogenesis and molecular cargos.

## 5. Conclusions

Pregnancy is associated with a significant increase in the number of total EVs and placenta-derived EVs in maternal circulation across gestation. Hypoxia and hyperglycemia are the main triggers for the release of placenta-derived EVs. Compelling evidence supports placental-associated complications such as preeclampsia and fetal growth restriction as having some alterations in the number and the contents of EVs. Therefore, changes in the profile of placenta-derived EVs are a very attractive tool for the identification of placental-associated complications in asymptomatic pregnant women. Placental hypoxia in women with Bart’s hydrops fetuses may exhibit changes in the numbers and/or functions of placental-derived EVs, and thus could serve as a liquid biopsy tool for an early non-invasive prenatal diagnosis. Nonetheless, standardization of the method for placenta-derived EV detection is needed in order to open a new avenue for an early identification, monitoring, and therapeutic intervention of women at risk for Bart’s hydrops fetalis and other placental-associated conditions.

## Figures and Tables

**Figure 1 ijms-24-05658-f001:**
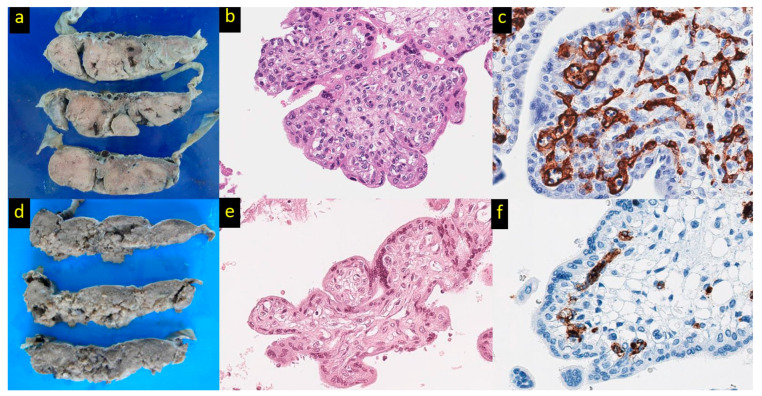
Placental pathology in Hb Bart’s disease. (**a**) The placenta belonging to Hb Bart’s disease shows enlarged, pale, and edematous cut surfaces in comparison to the placenta of non-Hb Bart’s disease (**d**). (**b**) Immature intermediate villi with a bulbous contour possessing conspicuous myofibroblasts at the periphery beneath the trophoblastic layer referred to as peripheral villous stromal hypercellularity in comparison to the control (**e**) (hematoxylin and eosin, original magnification ×400). (**c**) A branching vascular pattern with increased vessel number and endothelial thickness in comparison to the control (**f**) (CD34 immunostaining, original magnification ×600).

**Figure 2 ijms-24-05658-f002:**
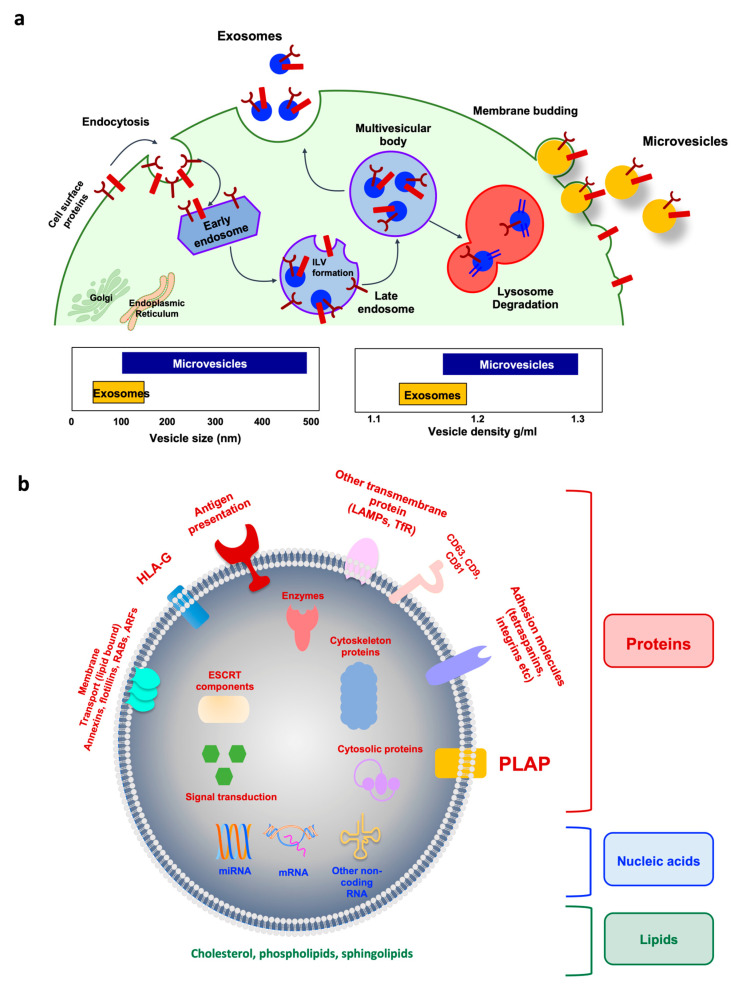
Extracellular vesicle biogenesis and compositions. (**a**) Exosome biogenesis and intracellular life are depicted on the left. Cell surface proteins are endocytosed and trafficked to early endosomes. Once sorted to late endosomes, the endosomal sorting complex required for transport of (ESCRT)-0 complex recruits ubiquitinated proteins, while ESCRT-I and -II mediate the budding of intraluminal vesicles (ILVs). The multivesicular body (MVB) can either follow a degradation pathway fusing with lysosomes or proceed to release the ILV contents (as exosomes) to the extracellular space by an exocytic step. Microvesicle biogenesis via plasma membrane budding is illustrated on the right. Transmembrane proteins are clustered in discrete membrane domains that promote outward membrane budding. Tetraspanins and other proteins abundant at the domain may have a role by promoting the sorting of other components. Lipid-anchored (myristoylation, palmitoylation) proteins accumulate proteins in the lumen as well as contribute to membrane curvature. Additional mechanisms of microvesicle formation include the calcium-activated scramblases, which randomize the distribution of lipids between the two faces of the plasma membrane. The cytoskeleton becomes looser, while cytosolic proteins and RNA molecules are sorted into microvesicles. The specific ATPase VPS4 mediates the disassembly of the spiral by pulling its end. (**b**) Representative structure of exosomes with cargos. Note that placenta alkaline phosphatase (PLAP) is a specific marker of placenta-derived EVs. ARFs, ADP ribosylation factors; CD, cluster of differentiation; ESCRT, endosomal sorting complex required for transport; LAMPs, lysosome-associated membrane glycoproteins; mRNA, messenger RNA; miRNA, microRNA; RABs, member of RAS superfamily of small G proteins; TfR, transferrin receptor.

**Figure 3 ijms-24-05658-f003:**
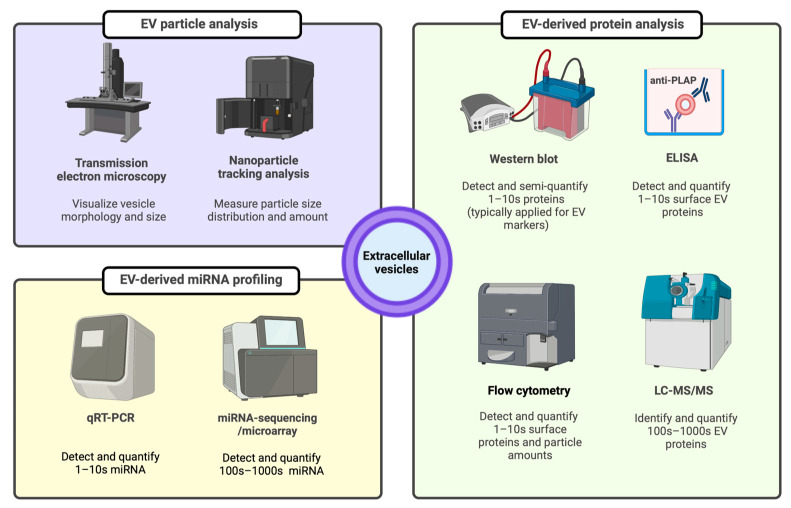
EV characterization methods for particle, protein, and miRNA evidence. ELISA: enzyme linked immunosorbent assay, EV: extracellular vesicle, LC-MS/MS: liquid chromatography-tandem mass spectrometry, miRNA: microRNA, PLAP: placental alkaline phosphatase protein, qRT-PCR: quantitative real-time polymerase chain reaction.

**Figure 4 ijms-24-05658-f004:**
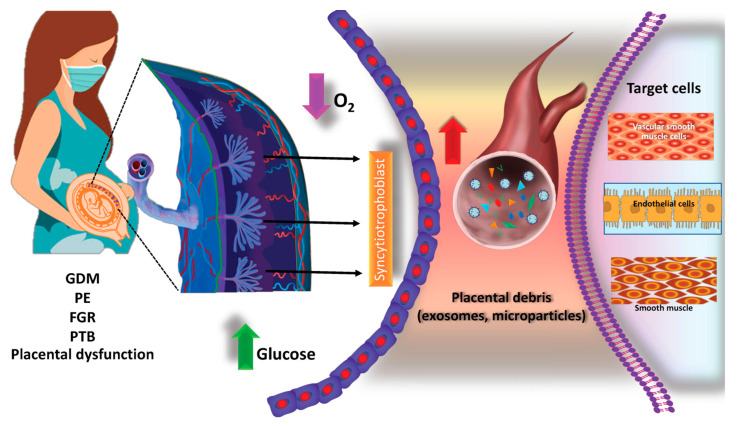
Potential roles of placental debris and exosome changes during pregnancy-related complications. Placental debris and EVs (including exosomes and microvesicles) released from the placenta under different micro-environment conditions (such as hypoxia or high glucose) and their targeting of neighboring cells in the placenta and distant organs such as skeletal muscles. EVs are released by placental cells (such as syncytiotrophoblasts, cytotrophoblasts (CTs), and extravillous trophoblasts (EVTs)) and other cells in the placenta such as the placental mesenchymal stem cells. EVs from the placenta can enter maternal circulation and target distant cells such as skeletal muscles, endothelial cells, or vascular smooth muscle cells. Placenta-derived EVs’ concentration, content, and bioactivity changes in pregnancy and pregnancy-related disorders including GDM, PE, PTB, and FGR. GDM: gestational diabetes, PE: preeclampsia; FGR: fetal growth restriction, PTB: preterm birth.

**Figure 5 ijms-24-05658-f005:**
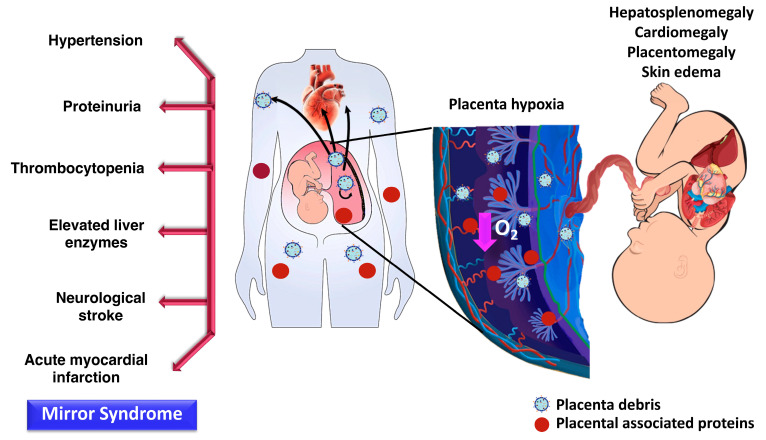
The possible roles of placenta-derived exosomes in the pathogenesis of Bart’s hydrops fetalis. Features of fetus with Bart’s hydrops are hepatosplenomegaly, cardiomegaly, ascites, pleural effusion, skin edema, and/or placentomegaly. Women with Bart’s hydropic fetuses can be presented with preeclampsia features called “mirror syndrome”. Placental hypoxia in Bart’s hydrops fetus is hypothesized to trigger the release of placental debris or placenta-derived exosomes which enter maternal circulation.

**Figure 6 ijms-24-05658-f006:**
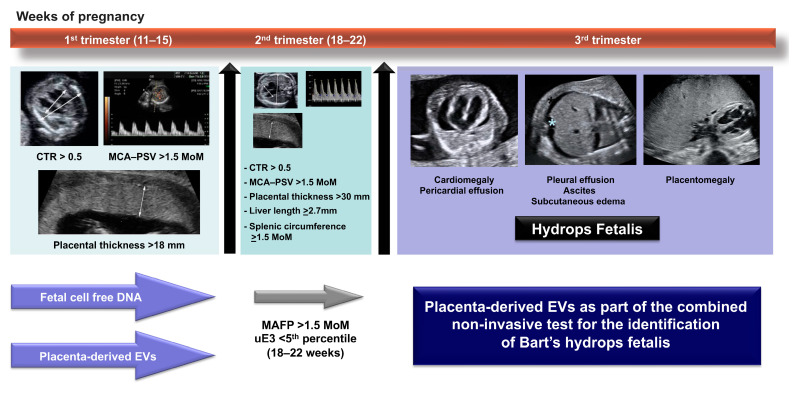
An integrative diagnostic model combining current biomarkers and placenta-derived EVs for the identification of Bart’s hydrops fetalis throughout gestation. CTR: cardiothoracic diameter ratio, MCA-PSV: middle cerebral artery peak systolic velocity, MoM: multiple of median, MAFP: maternal alpha fetoprotein, uE3: unconjugated estriol.

**Table 1 ijms-24-05658-t001:** Performance of Sonographic Markers in Predicting Fetal Hb Bart’s Disease.

	Study	Gestational Age (Week)	Cut-Off Value	Sensitivity (%)	Specificity (%)	References
Potential biomarkers	Cardio-thoracic diameter ratio	12–15	≥0.50	75–100	90–100	[18,46,47,48,49,50,59]
		18–22	≥0.52	88–100	87–96	
	Middle cerebral artery—peak systolic velocity	12–15	≥1.5 MoM	17–56	79–96	[48,49,53]
		16–22	≥1.5 MoM	64–85	98–100	
	Placental thickness	12–15	≥18 mm	72.9	68.8	[48,49,52]
		18–22	≥30 mm	74–100	96.1	
Other additional biomarkers	Thickened nuchal translucency	11–14	>95th percentile	16.7	98.6	[49]
	Cardio-biparietal diameter ratio	17–22	≥0.45	84–91	77–93	[60,61]
	Cardiac circumference	15–22	1.17 MoM	86.4	78.1	[57]
	Global sphericity index	18–22	1.17	74.1	88.2	[62]
	Liver length	18–22	≥27.0 mm	71.3	95.5	[54]
	Splenic circumference	18–22	≥1.5 MoM	70.1	83.0	[63]
	Splenic artery—peak systolic velocity	18–22	≥1.51 MoM	84.4	98.1	[55]

Abbreviation: MoM, multiples of the median.

**Table 2 ijms-24-05658-t002:** Performance of Maternal Biomarkers in Predicting Fetal Hb Bart’s Disease.

Specimens/Procedures	Biomarkers	Cut-Off Value	Test Performance	References
Fetal cell-free DNA	Fetal cell-free DNA (GA 11–13 weeks)	-	sens: 98.08% spec: 96.06%	[6]
Maternal serum biomarker in second trimester	MAFP	≥1.5 MoM	sens: 87.2% spec: 74.5%	[23,24,25,27,64]
	uE3	<5th percentile	*p* < 0.001	
	Free β-hCG	Increased	*p* = 0.543	
	PAPP-A	Increased	*p* = 0.777	
	Inhibin-A	Increased	*p* = 0.001	
	PlGF	Increased	*p* = 0.008	
	sFlt-1	Increased	*p* = 0.139	
	sFlt-1/PlGF ratio	Decreased	*p* = 0.001	
Combined biochemical and imaging markers	Predictive model (MAFP + uE3) 1/1 + e^−[2.876 + 1.333(AFP_MoM) − 6.310(uE3_MoM)]^	0.5	sens: 61.5% spec: 98.1%	[64]
	MAFP + MCA-PSV (>1.5 MoM)	-	sens: 97.9% spec: 69.1%	
	MAFP + CTR (>0.5)	-	sens: 100.0% spec: 59.3%	
	MAFP + Placental thickness (>3.0 cm)	-	sens: 88.9% spec: 69.5%	
	MAFP + MCA-PSV + CTR	-	sens: 100.0% spec: 48.1%	

Abbreviations: β-hCG, human chorionic gonadotropin beta-subunit; CTR, cardiac diameter/thoracic diameter ratio; GA, gestational age; MAFP, maternal alpha fetoprotein; MCA-PSV, peak systolic velocity in the fetal middle cerebral artery; MoM, multiples of the median; PAPP-A, pregnancy associated plasma protein-A; PlGF, placental growth factor; sens, sensitivity; sFlt-1, soluble fms-like tyrosine kinase-1; spec, specificity; uE3: unconjugated estriol.

**Table 3 ijms-24-05658-t003:** Advantages and disadvantages of different biomarkers for the identification of Hb Bart’s hydrops fetalis.

	Advantages	Disadvantages
Sonographic markers	Non-invasive technique High predictive performance	Require specialized sonographers Require specific equipment Operator-dependent
Biochemical markers	Non-invasive technique Can perform simultaneously with second trimester Down syndrome screening	Poor predictive performance Require validation Other conditions such as infection can interfere with the result
Cell-free fetal DNA	Non-invasive technique High predictive performance Can perform simultaneously with Down syndrome screening	Relatively high costs Complex techniques and laboratory device Requires further validation. Thus, the search for other biomarkers is continuing
Extracellular vesicles	Non-invasive technique Contain molecular information of dynamic cell and tissue states of disease pathophysiological changes and complications Compatible with multiple etiologies of hydrop fetalis	Relatively high costs Complex techniques and laboratory device Predictive performance requires further studies

**Table 4 ijms-24-05658-t004:** EV classification and their biophysical characteristics.

EV Subpopulation	Diameter	Sedimentation in Centrifugation	Molecular Markers	Cellular Origin
Exosomes	40–150 nm	100,000–200,000× *g* for 1–2 h	CD9, CD63, CD81, TSG101, Alix	Multivesicular bodies
Microvesicles (or ectosomes)	100–1000 nm	10,000–20,000× *g* for 10–20 min	Integrins, selectins, CD40 ligand	Plasma membrane budding
Apoptotic bodies (or apoptotic cell-derived EVs)	1–5 µm	1000–2000× *g* for 10–20 min	Histones, Annexin V	Apoptotic cell membrane blebbing

## Data Availability

All data have been presented in the manuscript.

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
