# Peer review of "Placenta-Derived Extracellular Vesicles in Pregnancy Complications and Prospects on a Liquid Biopsy for Hemoglobin Bart’s Disease"

_ijms, 2023, doi:10.3390/ijms24065658_

Round 1

Reviewer 1 Report

1.    In the table 1. and table 2, the authors list several quite established practice for sonographic diagnosis of Hb Bart’s disease, I think it would be helpful to compare side by side the pros and cons of this method and the potential EV related method.  Otherwise, these two tables seem divergent from the main purpose of this paper.

2.    Overall, the figures presented in the manuscript are not with high quality and not clear. For example, figure 1. For the experienced researchers in the field, people might be able to identify the normal look of Placenta instantly/easily, or the disease associated one. While the review is also indented to a broad range of readers from different fields, The authors should have control images in Figure 1a, 1b, 1c.  Meanwhile, are these images previously published, if so, what is the source, and are they allowed to use here. The authors should be careful about it.

3.    Figure 2 a, the authors should label the cartoon used in the figure,

4.    Figure 5, remove the unrecognizable symbols.

Reviewer 2 Report

The review by Chaemsaithong et al. suggests using the placenta-derived extracellular vesicles as a potential method for diagnosing placental complications focusing on Bart’s hydrop fetalis. The possibility of identifying and monitoring Bart’s hydrops fetalis throughout gestation using a non-invasive test is highly desirable. Still, after reading the article, I feel that the hypothesis to use this approach is too far. Indeed, the characterization of placenta-derived extracellular vesicles related to Bart’s hydrop fetalis is in a very early stage. My suggestion is to enlarge the paper's focus on the correlation between placenta-derived extracellular vesicles and placental complications. Indeed, section 3 is informative and clear. Of course, this means a profound reorganization of the paper and the amendment of the title.

Minor revision

Pg 1 line 26 “Placenta-derived EVs an be detected” please rephrase

Pg 7 lines 253-256 ” I suggest better contextualizing the role of cell trafficking between the mother and the fetus in EVs scenario

Pg 7 lines 361-363 Like other cells, “shed syncytiotrophoblast microvilli” can be detected in maternal plasma and is thought to contribute to fetal immune tolerance” I suggest better contextualizing the sentence in the EVs scenario

Reviewer 3 Report

In their large review the authors summarize Bart’s disease and alpha-Thalassaemia, and how these can be detected during pregnancy. They discuss in detail fetus-derived extracellular vesicles and how they may be used in the future for such detections. Although this review can only speculate on the clinical potential, and the reviewer remains skeptical, this sure can be seen as a legitimate research proposal, which should be tested. Besides some passages at the end the paper is well written, figures and tables support the text. Perhaps a final check of font size or readability of figure text, or rearrangement of figures should be done by the editors (e.g. 2a and b, suggest: a above b, instead of side-by-side).

Round 2

Reviewer 2 Report

This reviewer suggests accepting the paper in its present form.

Author Response

Reviewer's comments

This reviewer suggests accepting the paper in its present form.

Reply: We appreciate that the reviewer recommends accepting the paper in its present form.